# Association of Antioxidant Vitamins A, C, E and Carotenoids with Cognitive Performance over Time: A Cohort Study of Middle-Aged Adults

**DOI:** 10.3390/nu12113558

**Published:** 2020-11-20

**Authors:** May A. Beydoun, Jose A. Canas, Marie T. Fanelli-Kuczmarski, Ana I. Maldonado, Danielle Shaked, Mika Kivimaki, Michele K. Evans, Alan B. Zonderman

**Affiliations:** 1Laboratory of Epidemiology and Population Sciences, National Institute on Aging, Intramural Research Program, NIA/NIH/IRP, Baltimore, MD 21224, USA; aim@umbc.edu (A.I.M.); EvansM@grc.nia.nih.gov (M.K.E.); zondermana@mail.nih.gov (A.B.Z.); 2Department of Pediatrics, Johns Hopkins Medical Institutions, Saint Petersburg, FL 33701, USA; jcanas1@jhmi.edu; 3Department of Behavioral Health and Nutrition, University of Delaware, Newark, DE 19716, USA; mfk@udel.edu; 4Department of Psychology, VA Boston Healthcare System, Boston, MA 02130, USA; danielle.shaked@va.gov; 5Department of Epidemiology and Public Health, University College London, London WC1E 6BT, UK; m.kivimaki@ucl.ac.uk

**Keywords:** dietary carotenoids, antioxidants, cognitive performance, longitudinal studies, urban adults

## Abstract

Carotenoids may strengthen the association of antioxidant vitamins A, C, and E with favorable cognitive outcomes over time, though a few prospective studies have examined this hypothesis. We evaluated the longitudinal data from 1251 participants in the Healthy Aging in Neighborhoods of Diversity across the Life Span (HANDLS) study (Age at visit 1 in 2004–2009 (v_1_): 30–65 years). Vitamins A, C, and E dietary intakes and total and individual dietary carotenoids were computed using two 24-h recalls at v_1_. Cognitive tests, covering global mental status and domains of memory/learning, attention, psychomotor speed, visuo-spatial, language/verbal, and executive function were conducted at v_1_ and/or v_2_ (2009–2013); mean ± SD follow-up: 4.66 ± 0.93 years. Mixed-effects linear regression models detected an interaction between vitamin E and total (and individual) carotenoids for three of 11 cognitive tests at v_1_, with only one meeting the statistical significance upon multiple testing correction whereby vitamin E was linked with greater verbal memory performance in the uppermost total carotenoid tertile (*γ*_0*a*_ = +0.26 ± 0.08, *p* = 0.002), a synergism largely driven by carotenoid lycopene. Vitamins A and C showed no consistent interactions with carotenoids. In conclusion, we provide partial evidence for synergism between vitamin E and carotenoids in relation to better baseline cognitive performance, pending further studies with time-dependent exposures and randomized trials directly examining this synergism.

## 1. Introduction

Cognitive impairment, across various levels of severity, is a major determinant of disability and long-term institutionalization in old age [1,2]. With the aging of populations over time, retaining cognitive function in old age is an increasingly important public health challenge. To extend the intact cognitive functioning into old age, and reduce the costs of care in late life, a greater focus on and understanding of factors that may decelerate or accelerate cognitive decline is needed [3,4]. One such factor is oxidative stress, which has received considerable attention over the past few decades in studies uncovering etiology, treatment, and preventive measures against age-related cognitive decline.

Oxidative stress is an imbalance between pro-oxidant and antioxidant states, which tilts more toward pro-oxidant metabolism [3]. Such metabolic stress may contribute to cognitive decline, neurodegenerative diseases such as Alzheimer’s disease (AD), and other age-related conditions such as cardiovascular disease and cancer [5,6,7]. The brain may be particularly vulnerable to reactive oxygen species (ROS) because of their high lipid and iron content and metabolism accounting for ~20% of all bodily oxygen consumption [8]. ROS exposure can trigger unfavorable DNA oxidative modifications within the brain that accumulate because of DNA repair impairment with age [9,10,11].

Dietary intake of antioxidants such as β-carotene (as well as other carotenoids), vitamins A, C, and E were shown to reduce lipid peroxidation [8], the production of ROS [12], apoptosis [12], and protein and DNA oxidative damage [13,14]. Vitamin A deficiency was also linked to iron deficiency anemia, a risk factor for cognitive impairment [15,16,17,18]. Epidemiological evidence from observational studies thus far suggests that those antioxidants in the diet, supplements, and serum may play a neuroprotective role [3,19,20,21,22,23]. Given that sub-clinical cognitive deterioration in older age may impact the dietary intakes at baseline, it is thus crucial to examine the associations between dietary antioxidant and cognitive outcomes, including change over time, in a relatively young and healthy population [24]. Consequently, dietary antioxidants may potentially slow the pace of cognitive decline among middle-aged adults, thus delaying the onset of AD and other forms of dementia in older age. Moreover, there is some evidence of biological synergism between some carotenoids (e.g., lycopene and lutein) and other antioxidants (e.g., vitamin E), and potential antagonism for others, whereby increased intake of carotenoid β-carotene may reduce blood levels of vitamin E [15]. To date, however, no studies have investigated whether carotenoids in general may interact with vitamins A, C, or E in relation to cognitive performance and decline.

In this report, we use the longitudinal data from a large bi-racial study of urban middle-aged adults, to examine adjusted associations among several dietary antioxidants and cognitive performance and change over time. Specifically, we examined the relationships of vitamins A, C, and E with cognitive performance and decline across the levels of dietary total carotenoid intake and tested interactions of vitamin A, C, and E with total and individual carotenoids, such as α-carotene, β-carotene, lutein + zeaxanthin, β-cryptoxanthin, and lycopene in relation to cognitive outcomes.

## 2. Materials and Methods

### 2.1. Database

The Healthy Aging in Neighborhoods of Diversity across the Life Span (HANDLS) study is an ongoing prospective cohort study initiated in 2004. The study’s primary objective is to examine health disparities in outcomes such as cardiovascular disease and cognitive aging. A special feature of HANDLS is its recruitment of a racially and socioeconomically diverse sample of African American and White urban-dwelling middle-aged adults. HANDLS collected measures of cognitive performance across time points, in addition to clinical and molecular biomarkers that cover several physiological systems. Primary sampling units in HANDLS consisted of thirteen neighborhoods in Baltimore, MD and an area probability sampling strategy was applied [25]. Participants provided written informed consent upon thorough review of a protocol booklet that was written in layman’s terms accompanied by a video detailing all procedures and potential future re-contacts. The HANDLS study was approved by the Institutional Review Board of the National Institutes of Health, National Institute of Environmental Health Sciences (09-AG-N248).

### 2.2. Study Population

The initial cohort of HANDLS consisted of 3720 participants (30–65 years, African American and White, Phase I, v_1_: 2004–2009), in which screening, recruitment, and a household interview that incorporated a 24-h dietary recall were completed. The next phase of the study was conducted in Medical Research Vehicles [MRVs] (also known as Phase II of v_1_: 2004–2009); in-depth examinations were carried out, which included a fasting blood draw, a complete physical examination, a dual-energy X-ray absorptiometry scan, an electrocardiogram, a cognitive assessment, and a second 24-h dietary recall. During the period of 2009–2013, a first follow-up visit (v_2_) was conducted and consisted primarily of an MRV in-depth examination visit that assessed the cognitive performance in a similar manner as in v_1_. Among the 2468 participants who completed v_2_, the mean ± SD follow-up time was estimated at 4.66 ± 0.93 years.

In this study, up to two repeats of cognitive test scores from v_1_ and/or v_2_ were used along with dietary exposures measured at v_1_ with two 24-h recalls, and covariates measured at v_1_ as well, with some (supplement use and employment) measured at v_2_. The participant flowchart is detailed in Figure 1. First, out of the initial sample (*N* = 3720, Sample 1a), we included participants who had complete dietary exposure at v1 (*N* = 2177, Sample 1b). Valid cognitive test scores were available for up to 2922 participants at v_1_; and 2348 at v_2_ (primarily excluding participants because of literacy-related or physical/sensory limitations), corresponding to up to 4812 observations. Out of the 2177 participants included in Sample 1b, complete and valid cognitive test scores were available for 1956–2115 participants, depending on the cognitive test, corresponding to 2982–3592 observations (Samples 2a–2k), with 1.6–1.7 observations/participant. At a third stage, individuals with missing data on all other covariates (measured mainly at v_1_, with the exception of supplement use and employment status measured at v_2_: See Section 2.6) were excluded, yielding final samples of 1210–1251 participants, depending on the cognitive test score outcome, with 2055–2447 observations and 1.9–2.0 observations/participant (Samples 3a–3k). The sub-sample of participants with complete data on all cognitive test scores simultaneously at v_1_ and/or v_2_, exposures and covariates had a significantly lower proportion of men compared to the remaining sample of participants who were excluded from the initial HANDLS cohort (40.2% (*N* = 1158) vs. 47.6% (*N* = 2568). No differences in age, sex, or race between the included and excluded participants were observed.

### 2.3. Dietary Assessment

Dietary factors included in this study were measured at v_1_. Two 24-h dietary recalls were administered at this visit, by utilizing the US Department of Agriculture (USDA) Automated Multiple Pass Method, a structured computerized interview [26]. Participants used measurement aids, which include measuring spoons, cups, ruler, as well as an illustrated Food Model Booklet to estimate the portion sizes consumed. Both recalls were administered by trained interviewers, in-person, one during the household interview and another at the MRV for v_1_ and one in-person interview at MRV with second recall by telephone for v_2_. First and second recalls were separated by 4–10 days. Survey Net was utilized by trained nutrition professionals who matched the consumed foods with 8-digit codes derived from the Food and Nutrient Database for Dietary Studies (FNDDS) version 3.0, Reference [27] and the food group My pyramid equivalents database [28]. Dietary carotenoids, vitamins A, C, and E were among the many nutrients available in the FNDDS, from which daily values could be computed, using the average from the two 24-h recalls conducted at visit 1.

The main stratifying variables of interest were dietary carotenoids (α-carotene, β-carotene, lutein + zeaxanthin, β-cryptoxanthin, and lycopene in μg/day) and summed up into “total carotenoids,” as was done in other studies [29]. Days 1 and 2 of visit 1 24-h recalls were averaged first for each carotenoid, then summed up to obtain “total carotenoid” intake. The same approach was applied to all other dietary variables, including the other antioxidant exposures: vitamins A (Retinol equivalents, RE/day), C (mg/day), and E (mg/day); total energy intake (kcal/day) and the *n*-3:*n*-6 polyunsaturated fatty acids (PUFA) ratio. *n*-3:*n*-6 PUFA was the ratio of LNA, DHA, *n*3-DPA, and EPA over LA, *n*-6 DPA, and AA. This dietary factor was previously linked to faster cognitive decline as well as increased levels of depressive symptoms [30,31]. The Healthy Eating Index (HEI-2010) total score was estimated using an algorithm applied to each day of recall. Then the two recall estimates of HEI-2010 total score were averaged to reflect overall dietary quality. Steps for calculating HEI-2010 are found in the following link: [32].

### 2.4. Cognitive Outcome Measures

Cognitive performance was measured with seven tests resulting in 11 test scores, covering seven primary cognitive domains: Global, attention, learning/memory, executive function, visuo-spatial/visuo-construction ability, psychomotor speed, language/verbal. The tests were the Mini-Mental State Examination (MMSE) [33], the California Verbal Learning Test (CVLT) immediate (List A) and Delayed Free Recall (DFR) [34], Digit Span Forward and Backwards tests (DS-F and DS-B) [35], the Benton Visual Retention Test (BVRT) [36], Animal Fluency test (AF) [37,38], Brief Test of Attention (BTA) [39], Trails A and B [40], and the Clock Drawing Test (CDT) [41] (Appendix A). All participants were deemed able to understand informed consent and the protocol. All participants completed the MMSE, a cognitive screener capturing global cognitive performance, but a comprehensive dementia evaluation was not conducted. On the MMSE, a score of ≥24, a widely accepted cut-off, was considered unimpaired. It appeared that a low MMSE score was a function of relatively low literacy level, as opposed to dementia. Nevertheless, individuals with impaired MMSE were excluded from the analysis. All cognitive scores are in the direction of higher score → better performance with the exception of BVRT (# of errors) and TRAILS A and B (# of sec. to complete).

### 2.5. Covariates

#### 2.5.1. Socio-Demographic and SES Covariates 

Covariates added in multivariable models were previously shown to be related either to the outcomes or the exposures, or both. Those included age at visit 1 (in years), sex, race (African-American vs. White), self-reported household income categorized as either <125% or ≥125% of the 2004 Health and Human Services poverty guidelines (termed poverty status) [42], completed years of education, re-categorized as “0–8: <High School (HS),” “9–12: HS,” “>12: >HS,” current employment status (Unemployed vs. Employed). Literacy was also included among SES covariates, as reflect by total score on the 3rd edition of the wide range achievement test (WRAT-3), described in Appendix A.

#### 2.5.2. Lifestyle and Health-Related Covariates

Other potential confounders can be classified as lifestyle or health-related covariates. Those include current tobacco smoking (“yes,” “missing” vs. “no”), current use of illicit drugs which include marijuana, cocaine, and opiates (“yes,” “missing” vs. “no”), measured body mass index (weight/height-squared, kg/m^2^), self-rated health (“poor or average,” “good” vs. “Very good or excellent”); Center for Epidemiological Studies-Depression score, based on the 20-item scale, potentially ranging between 0 and 60, with higher score reflective greater depressive symptoms (Appendix A); history or diagnosis or medications for diabetes (No vs. pre-diabetic vs. diabetic) and for hypertension; history of dyslipidemia and/or use of statins (yes vs. no); history of cardiovascular disease (atrial fibrillation, angina, coronary artery disease, congestive heart failure, or myocardial infarction). Supplement use was measured only at visit 2. However, length of use of supplements containing a specific micronutrient could be derived allowing the extrapolation of use into the first visit of HANDLS. Thus, participants could be categorized as: “non-users”, “users at visits 1 and 2”, and “users at visit 2 only”. In this study, supplements containing vitamins A, C, and E and various carotenoids were of interest. The length of use for each was averaged, and this average length was then determined to be prior to v_1_ or during the period between v_1_ and v_2_. All other participants were considered non-users at visits 1 or 2. Details related to supplement use assessment in HANDLS are provided elsewhere [43].

### 2.6. Data Handling and Statistical Analysis

Analyses were completed with STATA Release 16 [44]. Description of key variable distributions were presented for the total sample and stratified by tertiles (T) of total carotenoids. Means of continuous variables across tertiles were compared using linear regression models, first to examine trends across tertiles, and then to contrast T_2_ vs. T_1_ and T_3_ vs. T_1_. Chi-square tests for independence were used to make distribution comparisons of categorical variables across tertiles of carotenoids. Multiple linear, logistic, and multinomial logit models were used to test those differences across carotenoid tertiles, while adjusting for age, sex, race, and poverty status. The analyses testing the main hypotheses consisted of several linear mixed-effects time interval regression models that were stratified by total carotenoid intake tertiles [45]. In each model, and for each stratum, outcomes included one of 11 cognitive test scores with up to two repeats at v_1_ and/or v_2_ and predictors were each of five individual carotenoids, total carotenoids, vitamins A, C, and E measured at v_1_. All models incorporated number of years elapsed between visits (TIME) and two-way interaction terms between exposure or covariates with TIME. The model adjusted v1 cognitive test scores as well as annualized change over time for potentially confounding covariates. These covariates (listed under Section 2.5) included other antioxidants and total carotenoids, socio-demographic, lifestyle, and health-related factors. Modeling was done in two steps. In model 1, minimal adjustment was made on the other two antioxidants, total carotenoids, age, sex, race, poverty status, education, literacy, employment status, and total caloric intake. In model 2, further adjustment was made on tobacco and drug use, HEI-2010 total score, BMI, self-rated health, CES-D total score, hypertension, diabetes, dyslipidemia (or statin use), cardiovascular disease, and antioxidant-containing supplement use. The two-way interaction terms exposure × TIME are interpreted as the effects of exposures (net of covariates) on the slope or annual rate of change in each cognitive test score. Main effects of carotenoids and antioxidants, also included in the mixed-effect linear regression models, allowed us to examine the net exposure effect on baseline cognitive performance, i.e., the cross-sectional exposure–outcome association, controlling for v_1_ covariates. The mean number of repeats/participant for each outcome measure ranged between 1.7 and 2.0, reflecting missingness of outcomes ranging between 0% and 15%. In each model, we assumed missingness of outcomes to be at random (Appendix A) [46].

Two- and three-way interaction terms between exposure, the effect modifier (i.e., total carotenoid tertile), and TIME were added in separate models to test for moderation by carotenoid tertile. Covariates under the section specified were incorporated in all models, except for the stratifying variable in the stratified models. Continuous covariates and exposure were centered at their means in mixed-effects regression models. In the main analysis, covariate adjustment was incremental (Model 1 vs. Model 2).

In all models, to account for sample selectivity, we applied a two-stage Heckman selection strategy to the mixed-effects linear regression models. At a first stage, each binary selection variable was regressed on sex, race, baseline age, and poverty status using a probit model from which an inverse mills ratio (IMR) was estimated, utilizing the conditional predicted selection probability. In a second stage, IMR was entered into the main causal mixed-effects regression models in a similar manner as other covariates [47].

Type I errors for each main effect and interaction term was set at 0.05 and 0.10, respectively [48], prior to multiple testing correction. A familywise Bonferroni approach was applied for this adjustment, accounting only for cognitive test score multiplicity. We thus assumed that each cognitive sub-scale score is considered a distinctive substantive hypothesis [49]. Thus, significance levels for main effects were adjusted to *p* < 0.0045 (0.05/11); 0.10/11 = 0.0091 for the two-way interaction terms, to 0.05 for three-way interaction terms similar to previously published research [50]. Multiple testing adjustment was mainly considered for the minimally adjusted model (i.e., Model 1). Model 2 was considered a sensitivity analysis for Model 1 that adjusted for additional covariates. In addition, other sensitivity analyses were conducted. One of these pertained only to MMSE, which was normalized using a method previously described [51]. All models with MMSE were then re-ran with this normalized measure as the outcome. The second sensitivity analysis examined interactions of vitamins A, C, and E with individual carotenoids in relation to cognitive performance and decline in the fully adjusted mixed-effects linear regression models, which additionally controlled total carotenoid intake.

## 3. Results

### 3.1. Characteristics of Study Participants by Total Carotenoid Tertiles

Study sample characteristics are presented in Table 1 across v_1_ dietary carotenoid tertiles. Upward linear trends across tertiles of carotenoids were observed for most dietary factors (i.e., individual carotenoids, vitamins A, C, and E, total caloric intake and HEI-2010). Moreover, the uppermost tertile of dietary carotenoids (T_3_) compared with the lowest tertile (T_1_) had a higher socio-economic status, in terms of education and employment status, as well as elevations in mean WRAT-3 score and proportion with very good/excellent health, while having a lower proportion of African-Americans, current tobacco users, individuals with hypertension and/or diabetes. Many of those associations remained statistically significant upon adjustment for socio-demographics variables (v_1_ age, sex, race, and poverty status). In terms of v_1_ cognitive test performance, when compared with T_1_, with a linear dose-response relationship, T_3_ total carotenoid intake was characterized by better performance on MMSE, BVRT, AF, and DS-B, even after adjusting for basic socio-demographics. Furthermore, annual rates of change in cognitive performance suggested an average improvement or a slower decline for T_3_ total carotenoid intake compared to T_1_, for MMSE, BVRT, BTA, AF, DS-F, and DS-B, even upon adjustment for the same socio-demographics listed above. 

### 3.2. Baseline and Change in Cognitive Performance vs. Individual/Total Carotenoids and Other Antioxidants: Mixed-Effects Regression Models

In the total population, and in the minimally adjusted model (Model 1) that included individual carotenoids as well as other antioxidants (Appendix A), higher vitamin E was associated with better baseline performance on CVLT-List A and with slower decline on TRAILS B performance (*p* < 0.05). Additionally, β-carotene and vitamin C were associated with faster decline on and poorer performance on CDT and DS-B, respectively. Notably, lycopene was linked to worse performance on CVLT-List A at v_1_. Nevertheless, none of these associations survived correction for multiple testing. Further adjustment for additional covariates (i.e., Model 2), attenuated many of those associations (data not shown), suggesting potential mediation through overall diet quality among others. Similar patterns were observed in models with total carotenoids.

### 3.3. Baseline and Change in Cognitive Performance vs. Interaction between Total Carotenoids and Other Antioxidants: Mixed-Effects Regression Models

The associations of dietary vitamins A, C and E with cognitive performance and decline over time across total carotenoid tertiles (T_1_ through T_3_) is presented in Table 2. Mixed-effects linear regression models that adjusted for minimal covariates (Model 1), excluding lifestyle and health-related factors, suggest that within T_3_ of total carotenoids, vitamin E intake was positively associated with a test of verbal learning and memory (CVLT-List A) at v_1_, reflecting immediate recall (*γ*_0*a*_ = +0.26 ± 0.08, *p* = 0.002, See Appendix A for notations). This association passed correction for multiple testing and remained statistically significant upon further adjustment for lifestyle and health-related factors, including HEI-2010 total score, a measure of overall dietary quality (*γ*_0*a*_ = +0.22 ± 0.09, *p* = 0.013). There was no evidence of heterogeneity, however, across tertiles of total carotenoid intake, with respect to the vitamin E vs. v_1_ CVLT-List A score association. In contrast, vitamin E was also positively associated with v_1_ DS-B score within T_3_ in Model 1 (*γ*_0*a*_ = +0.047 ± 0.022, *p* = 0.032), with a significantly stronger association within that tertile compared to T_1_ (*p* < 0.10 for the interaction term vitamin E by carotenoid tertile: T_3_ vs. T_1_). This relationship, however, within T_3_, did not survive multiple testing correction or further covariate adjustment (Model 2). Within T_2_ of total carotenoid intake, vitamin E intake was linked to slower decline on a test of executive function, namely TRAILS B (*γ*_1*a*_ = −1.25 ± 0.46, *p* = 0.007). This association passed multiple testing correction in Model 1 and remained statistically significant in Model 2 (*γ*_1*a*_ = −1.19 ± 0.50, *p* = 0.017), without evidence of heterogeneity with T_1_. With respect to vitamin A, a synergistic interaction was detected, whereby this vitamin was associated with faster decline on a test of visual memory and visuo-constructive ability (BVRT), but only within the uppermost tertile of carotenoid intake (T_3_); (*γ*_1*a*_ = +0.28 ± 0.06, *p* < 0.001). This association passed correction for multiple testing, retained statistical significance in Model 2 (*γ*_1*a*_ = +0.27 ± 0.07, *p* < 0.001) and indicated a large degree of heterogeneity when compared with the T_1_ vitamin A vs. BVRT change association. There were no distinctive patterns observed for the association of vitamin C with v_1_ cognitive performance or change across total carotenoid intake tertiles.

### 3.4. Baseline and Change in Cognitive Performance vs. Interaction between Individual Carotenoids and Other Antioxidants: Mixed-Effects Regression Models

When examining the interactions between individual carotenoid intake and that of each vitamins A, C, and E in relation to cognition (at v1 or change; Table 3), by running a series of fully adjusted linear mixed-effects models, vitamin C showed a consistent deleterious interaction with three (α-carotene, β-carotene and lutein + zeaxanthin) of five carotenoids with respect to change in CDT over time, a measure of visuo-spatial abilities, with higher intake of those carotenoids coupled with higher vitamin C intake increasing the pace of decline. This pattern was also detected with respect to the interaction between vitamin C and β-cryptoxanthin in relation to over time change in MMSE total score. This is in contrast to the interactions between vitamin C and 2 carotenoids (α- and β-carotene) which were linked to better performance on verbal memory as measured by CVLT-List A; as was the case between vitamin C and α-carotene which acted synergistically in relation to better performance on TRAILS B, a measure of executive function among others.

The synergistic interaction between vitamin A and carotenoids with respect to decline on BVRT over time was specific to β-carotene and lutein + zeaxanthin (*p* < 0.001 for interaction term between carotenoid, vitamin A and TIME); and was replicated for CDT with respect to β-carotene and β-cryptoxanthin. However, vitamin A interacted with α- and β-carotene yielding better v_1_ cognitive performance on CVLT-List A, AF, and TRAILS B (α-carotene) as well as better v_1_ cognitive performance on BVRT (β-carotene). Additionally, vitamin A interacted with other carotenoids yielding slower decline on TRAILS B (lutein + zeaxanthin) and DS-F (β-cryptoxanthin). Thus, interaction between vitamin A and carotenoids was consistent for v_1_ cognitive performance in terms of directionality, suggesting a putative protective effect, despite its inconsistency with respect to the cognitive change outcome.

In contrast, the interaction between vitamin E and individual carotenoids indicated that vitamin E was consistently associated with better v_1_ performance at higher levels of α-, β-carotene, and lycopene (or vice versa), specifically for AF, DS-F, and TRAILS B (α-carotene), DS-B (both α- and β-carotene; with *p* < 0.001 for α-carotene) and CVLT-List A (lycopene). The latter finding suggests that the interaction between total carotenoid tertile and vitamin E with respect to verbal memory is largely driven by lycopene intake. Findings of normalized MMSE scores were comparable to the originally scaled scores, with respect to tertile-specific findings as well as interactions between carotenoids and other antioxidants. 

## 4. Discussion

To our knowledge, this study is the first to examine the potential synergistic associations of carotenoids and other antioxidants in relation to cognitive performance and change in a prospective cohort study of a large bi-racial sample of urban middle-aged adults. Among the key findings, we detected consistent synergism between total (and individual) carotenoids and vitamin E in relation to baseline cognitive performance in domains of verbal memory, verbal fluency, attention, working memory, and executive function (See Table 2 and Table 3). Among those associations, vitamin E was associated with greater verbal memory performance in the uppermost tertile of carotenoids, an association largely driven by lycopene intake. Moreover, within the uppermost tertile of total carotenoids, vitamin A was associated with faster decline on a test of visual memory and visuo-constructive ability, an association that was driven by β-carotene and lutein + zeaxanthin. Nevertheless, both vitamins A and C showed inconsistent interactions with carotenoids with respect to both baseline cognitive performance and change in cognitive test scores over time. While no antagonism was found between β-carotene and vitamin E, there was some evidence of synergism between α-carotene and vitamin E in relation to performance in multiple domains of cognition. It is worth noting that within the total population, and after correction for multiple testing, there was limited evidence of an association between each of these antioxidants and cognitive performance or decline.

### 4.1. Previous Studies

There are several prospective cohort studies on the negative association between dietary antioxidants and the risk of dementia [52,53], while other studies suggested that dietary antioxidants, including vitamins C, E, and β-carotene were not associated with reduced AD risk [54,55]. Similarly, largely protective findings were reported when cognitive decline or incident cognitive impairment were among the main outcomes of interest and the exposures were dietary intakes of antioxidants [56,57,58], as was the case for long-term intake of carotenoids in relation to follow-up subjective cognitive function among women [23]. In another study, when both types of outcomes were considered, it was concluded that certain forms of tocopherols not found in dietary supplements but found only in foods may be at play [59]. This observation was corroborated by at least one recent study [60]. Similarly, Laurin and colleagues [55] found no association between midlife dietary intake of vitamins E and C and dementia incidence. At least four other cohort studies came to a similar conclusion with respect to those two dietary antioxidants [61,62,63,64]. Nevertheless, Jama et al. found that carotenoids, particularly β-carotene intake, may have beneficial effects on various cognitive outcomes, Reference [62] though others failed to detect such an association [56,63,64]. Numerous cross-sectional studies have also found that dietary carotenoids [65,66], vitamins A [67], C [65,68,69,70], and E [67,71] were associated with better cognitive performance, while others could not detect such associations [63,72]. In this study, the most robust findings included the potential protective effect of vitamin E from the diet with respect to verbal memory at one point in time, and its synergism with total carotenoid intake, particularly dietary lycopene, which, by itself, was shown to have a potentially adverse effect on verbal memory. Among the individual carotenoids, lycopene in plasma, directly reflecting dietary intakes, has generally been linked to better cognitive performance and reduced rate of cognitive decline [23,73,74,75].

Longitudinal studies that examined the relationships of various supplemental antioxidants with various cognitive outcomes found conflicting results: while vitamin C supplement use was related to lower AD risk in one cohort study, Reference [76] combined supplementation of vitamin E and vitamin C was associated with reduced prevalence and incidence of AD or other dementias and cognitive decline in four other cohort studies [77,78,79,80,81], whereas another study found this effect to be specific to Vitamin E supplements [82]. The findings demonstrating a protective effect of supplemental antioxidant use against cognitive impairment and decline was replicated in a large cohort study [83]. However, there was only borderline or little evidence of a cognitive benefit from use of antioxidant supplements, particularly vitamins C and E, according to at least six independent cohort studies [54,60,84,85,86,87], and several cross-sectional studies [61,84]. Nevertheless, at least a few randomized controlled trials (RCTs) have indicated a benefit from supplementation with vitamin C [88], particularly among non-smokers [89], or from a combination of various antioxidants (including vitamins C, E, and β-carotene) [90,91], while other RCTs that focused on vitamin E supplementation failed to do so [92,93,94]. In our study, adjustment for supplemental intake of antioxidants did not alter the key findings with dietary antioxidants as the main exposures.

Irrespective of the source of antioxidants, plasma concentration may be a good biomarker for oxidative stress status. In particular, an inverse association between plasma vitamin E among others and poor cognitive outcomes was found in at least five cross-sectional studies [95,96,97,98,99] and three cohort studies [100,101,102]. There was an evidence of a U-shaped association between sub-types of tocopherols in blood and cognitive impairment [101]. Moreover, among studies that examined the influence of plasma carotenoids, eleven detected a significant potential protective effect against cognitive impairment [19,22,73,74,75,97,103,104,105,106]. While generally results are mixed, they suggest that at least one antioxidant has a protective effect against the adverse cognitive outcomes. The present study, despite not including plasma concentrations of antioxidants among exposures, took into consideration the supplemental intake when looking at food-based sources of antioxidants.

### 4.2. Biological Mechanisms

Several findings suggest that oxidative stress characterized as an imbalance between the formation of reactive species and depletion of antioxidative defenses plays an important role in the pathogenesis of AD. First, the brains of AD patients have lesions that are characterized by the combined presence of two classes of abnormal structures, extracellular amyloid plaques and intraneuronal neurofibrillary tangles, both of which comprise highly insoluble, densely packed filaments associated with exposure to free radicals. These lesions are present in the brains of AD patients that are typically associated with free radical attacks (e.g., damage to DNA, protein oxidation, lipid peroxidation, and advanced glycosylation end products), and metals (e.g., iron, copper, zinc, and aluminum). Second, in vitro studies suggest that plant-based exogenous antioxidants (e.g., vitamins E and C, Ginkgo biloba extract EGb 761, melatonin, anthocyanins, flavonoids, and carotenoids) may reduce the toxicity of Aβ amyloids in the brains of AD patients and animal studies [107,108,109]. Based on these findings, it may be hypothesized that plant-based dietary antioxidants may help reduce the risk of AD. Vitamin E has not only antioxidant activity but also functions in other independent roles such as inhibiting brain protein kinase C activity [110]. This ability is most likely attributed to the multiple isoforms of Vitamin E [111,112]. It is recognized that the diet contains several Vitamin E isoforms while the results of this study reflect the intake of only one form of Vitamin E, α-tocopherol. Because of the increased use of oils in the US diet, there has been an increase in gamma tocopherol, a form that has similar antioxidant capacity but greater anti-inflammatory properties compared to α-tocopherol. Morris reported that only α and γ-tocopherols found in foods were linked to slower rate of cognitive decline over a 6-year period [59]. Third, carotenoids and their retinoid conversion products are potential therapeutic targets for AD treatment because of their capability to affect plaque formation, cholinergic transmission, ApoE and ABCA1 expressions, cholesterol content in the gut microbiota, and the pro-inflammatory environment of the brain [113]. As proof of concept, dietary supplementation with 2% and 4% date palm fruits reduced the cognitive and behavioral deficits in a transgenic mouse model for AD (amyloid precursor protein [APPsw]/Tg2576) offering neuroprotective effects [114].

### 4.3. Strengths and Limitations

Our study has notable strengths. First, it is among the largest prospective cohort studies examining the primary question of interest, making use of several cognitive function tests, advanced statistical techniques such as multiple linear mixed-effects regression models and Heckman selection, among others. Second, it is among a few studies to utilize data from two 24-h recalls as an estimate of usual dietary intakes of antioxidants. Our findings can also be extrapolated to several African–American and White populations, as HANDLS is representative of 14 urban settings across the United States.

Our study also has limitations. First, because of lack of factorial invariance by race, sex, or poverty status groups, the use of cognitive domains—obtained from confirmatory factor analysis—that were comparable between those groups was not feasible in this study. Data on supplemental intakes of antioxidants were not available for the baseline wave and thus were not accounted for in estimating the total intakes. However, a proxy measure was used, which coupled supplement use at follow-up visit with the length of use. In 2003–2006, more than half of adults are estimated to be dietary supplement users [115]. Moreover, supplement users also had higher intakes of vitamins A, C, and E from foods and in total than non-users [116], consequently exceeding tolerable upper limits for vitamins A and C, when compared to non-users [116]. Plasma concentration of antioxidants, reflecting food and supplemental sources, would be a more sensitive indicator of antioxidant and oxidative stress status, while reducing reporting bias [71]. Nevertheless, several large studies of healthy adults have shown that dietary values of each of those antioxidants and carotenoids represented in our study measured with 2 24-h recalls had moderate correlation with their respective plasma biomarker values [117,118]. Although, the baseline HANDLS study did not incorporate these measures, future studies of comparable low-income populations may support our analysis.

## 5. Conclusions

In sum, pending other longitudinal studies, we found limited evidence of an association between several antioxidants and cognitive performance or decline. Nevertheless, there was some evidence of consistent synergism between vitamin E and carotenoids in relation to better baseline cognitive performance, spanning several domains of cognition. The clinical interpretation and implications of this study are that a diet rich in vitamin E and carotenoids, including lycopene, may reduce the likelihood of cognitive decrements in the short term, particularly in the domain of verbal memory. Time-dependent changes in dietary and plasma levels of antioxidants need to be linked with changes in cognitive performance over time in future cohort studies, in a lagged manner, to ensure temporality of the associations and to incorporate supplemental sources of antioxidants. Pending such studies, randomized trials examining synergism between carotenoids and vitamin E in relation to cognitive performance and decline are needed.

## Figures and Tables

**Figure 1 nutrients-12-03558-f001:**
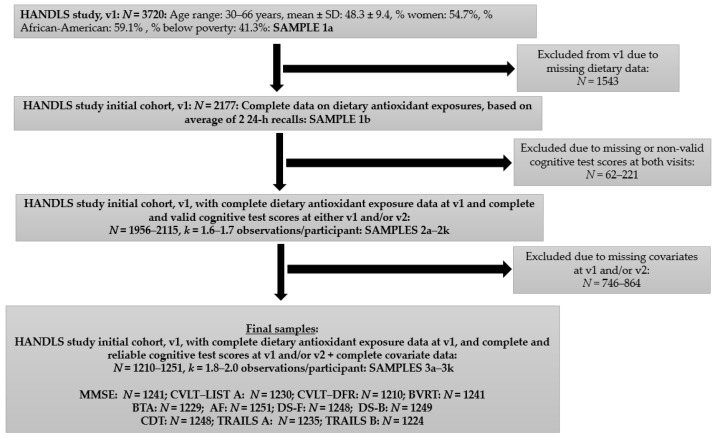
Participant flowchart. Abbreviations: AF = Animal Fluency; BTA = Brief Test of Attention; BVRT = Benton Visual Retention Test; CDT = Clock Drawing Test; CVLT-DFR = California Verbal Learning Test-Delayed Free Recall; CVLT-List A = California Verbal Learning Test-List A; DS-B = Digits Span-Backward; DS-F = Digits Span-Forward; HANDLS = Healthy Aging in Neighborhood of Diversity across the Lifespan; *k* = mean; MMSE = Mini-Mental State Examination; *N* = Number of participants; TRAILS A = Trailmaking Test, part A; TRAILS B = Trailmaking Test, part B.

**Table 1 nutrients-12-03558-t001:** Study sample characteristics by tertile of dietary total carotenoids for sub-sample with complete and valid MMSE data, HANDLS 2004–2013 ^a^.

	Overall	By Dietary Total Carotenoids Tertiles (μg/day) ^a^			
		T_1_	T_2_	T_3_	*p* ^b^
	(*X* ± SD), %	(*X* ± SD), %	(*X* ± SD), %	(*X* ± SD), %	
	(*N* = 1241)	(*N* = 402)	(*N* = 438)	(*N* = 401)	
Dietary carotenoids and other antioxidants at v_1_					
Dietary total carotenoids, μg/day	7162 ± 8261	1158 ± 686	4651 ± 1535 ***^,f^	15,923 ± 9404 ***^,f^	<0.001
α-carotene, μg/day	307 ± 727	45 ± 82	259 ± 373 ***^,f^	620 ± 1143 ***^,f^	<0.001
β-carotene, μg/day	1819 ± 2882	309 ± 289	1226 ± 987 ***^,f^	3980 ± 4152 ***^,f^	<0.001
Lutein + zeaxanthin, μg/day	1567 ± 3143	426 ± 297	967 ± 779 **^,f^	3366 ± 4994 ***^,f^	<0.001
β-cryptoxanthin, μg/day	104 ± 181	58 ± 109	124 ± 209 ***^,f^	127 ± 196 ***^,f^	<0.001
Lycopene, μg/day	3366 ± 5793	319 ± 470	2076 ± 1783 ***^,f^	7828 ± 8319 ***^,f^	<0.001
Dietary vitamin A, RE/day	589 ± 921	427 ± 821	554 ± 1108 *^,f^	790 ± 737 ***^,f^	<0.001
Dietary vitamin C, mg/day	73 ± 77	43 ± 47	80 ± 83 ***^,f^	96 ± 84 ***^,f^	<0.001
Dietary vitamin E, mg/day	6.48 ± 4.62	4.64 ± 3.32	6.39 ± 4.26 ***^,f^	8.46 ± 5.30 ***^,f^	<0.001
Baseline socio-demographic, SES and health-related variables					
Sex, % male	40.6	39.8	38.1	44.1	0.19
Age at v_1_, years	48.6 ± 8.9	48.8 ± 8.8	48.5 ± 9.0	48.5 ± 8.8	0.58
African-American, %	60.4	64.9	59.8	56.4 *^,f^	0.044
Poverty status, % (<125% PIR)	41.3	45.5	38.6 *	39.9	0.10
Education, yrs. Completed, %					<0.001
<HS	6.3	9.5	6.2	3.0	
HS	57.9	64.4	57.3	51.9 **^,f^	
>HS	35.9	26.1	36.5 **^,f^	45.1 ***^,f^	
Literacy, WRAT-3 score	42.6 ± 7.7	41.6 ± 7.7	42.3 ± 7.8	43.9 ± 7.3 ***^,f^	<0.001
% Employed in last month, yes	39.1	32.6	39.5	45.1 ***^,f^	0.001
% Employed in last month, missing	23.7	27.1	20.8 *	23.4	0.096
Baseline drug and tobacco use					
Any drug, current user, %	17.2	17.2	16.7	17.7	0.92
Any drug, missing, %	5.4	5.5	4.1	6.7	0.24
Tobacco, current user, %	44.7	55.2	42.5 ***^,f^	36.7 ***^,f^	<0.001
Tobacco, missing, %	0.9	0.8	0.5	1.5	0.26
Body mass index, kg/m^2^	30.0 ± 7.6	30.1 ± 7.6	30.4 ± 8.3	29.5 ± 6.9	0.27
Self-rated health					0.047
Poor/Average, %	23.8	27.1	25.1	19.0	
Good, %	41.1	41.5	39.3	42.6	
Very good/Excellent, %	35.1	31.3	35.6 **^,f^	38.4 ***^,f^	
HEI-2010 total score at v_1_	43.1 ± 11.8	39.2 ± 10.1	43.5 ± 11.6 ***^,f^	46.5 ± 12.4 ***^,f^	<0.001
Total energy intake, kcal/day	1993 ± 911	1763 ± 732	2010 ± 859 ***^,f^	2205 ± 1065 ***^,f^	<0.001
*N*3:*N*6 PUFA ratio	0.12 ± 0.09	0.12 ± 0.13	0.11 ± 0.04	0.12 ± 0.06	0.96
CES-D total score	14.3 ± 11.1	15.7 ± 12.0	14.1 ± 10.6 *	13.2 ± 10.5 **^,f^	0.001
Hypertension ^c^, %	46.0	50.8	45.4	41.9 *^,f^	0.040
Diabetes ^c^, %					0.70
No	67.6	66.4	66.2	70.3	
Pre-diabetic	16.0	16.2	17.1	14.5	
Diabetic	16.4	17.4	16.7	15.2	
Dyslipidemia ^c^, %	26.9	27.4	28.8	24.4	0.36
Cardiovascular disease ^c^, %	17.4	18.7	17.4	16.2	0.66
Supplement use, visit 2, %					0.12
Non-user	67.0	71.4	63.7	65.8	
At v_1_ and v_2_	1.9	2.2	1.6	1.8	
At v_2_ only	31.3	26.4	14.7 *^,f^	32.4	
Cognitive performance at v_1_, unadjusted ^d^					
MMSE	(*N* = 1239)	(*N* = 402)	(*N* = 437)	(*N* = 400)	<0.001
	27.8 ± 2.2	27.5 ± 2.3	27.7 ± 2.2	28.0 ± 1.9 ***^,f^	
CVLT-List A	(*N* = 1047)	(*N* = 345)	(*N* = 374)	(*N* = 328)	0.20
	24.5 ± 6.6	23.9 ± 6.5	24.8 ± 6.5	24.6 ± 6.7	
CVLT-DFR	(*N* = 1022)	(*N* = 334)	(*N* = 366)	(*N* = 322)	0.27
	7.3 ± 0.0	7.2 ± 2.8	7.2 ± 3.2	7.5 ± 3.0	
BVRT	(*N* = 1240)	(*N* = 405)	(*N* = 436)	(*N* = 399)	0.002
	6.3 ± 5.2	6.9 ± 5.5	6.3 ± 5.2	5.7 ± 4.9 **^,f^	
BTA	(*N* = 1083)	(*N* = 352)	(*N* = 380)	(*N* = 351)	0.012
	6.6 ± 2.2	6.4 ± 2.2	6.6 ± 2.2	6.9 ± 2.1 *	
AF	(*N* = 1235)	(*N* = 400)	(*N* = 431)	(*N* = 404)	0.006
	18.8 ± 5.5	18.5 ± 5.1	18.5 ± 5.5	19.5 ± 5.8 **^,f^	
DS-F	(*N* = 1222)	(*N* = 394)	(*N* = 427)	(*N* = 401)	0.016
	7.3 ± 2.2	7.1 ± 2.2	7.3 ± 2.2	7.5 ± 2.2 *	
DS-B	(*N* = 1212)	(*N* = 386)	(*N* = 428)	(*N* = 398)	0.004
	5.7 ± 2.2	5.4 ± 2.1	5.7 ± 2.2	5.9 ± 2.2 **^,f^	
CDT	(*N* = 1245)	(*N* = 404)	(*N* = 438)	(*N* = 403)	0.017
	8.8 ± 1.2	8.7 ± 1.3	8.8 ± 1.3	8.9 ± 1.2 *	
TRAILS A	(*N* = 1225)	(*N* = 393)	(*N* = 432)	(*N* = 400)	0.89
	37.5 ± 39.9	37.2 ± 32.4	38.3 ± 43.6	36.8 ± 42.5	
TRAILS B	(*N* = 1214)	(*N* = 388)	(*N* = 430)	(*N* = 396)	0.33
	145.6 ± 154.0	149.1 ± 152.1	149.0 ± 154.1	138.4 ± 156.0	
Annual rate of cognitive change, unadjusted ^e^					
MMSE	(*N* = 1241)	(*N* = 402)	(*N* = 438)	(*N* = 401)	<0.001
	+0.13 ± 1.70	−0.08 ± 1.78	+0.12 ± 1.75	+0.36 ± 1.50 ***^,f^	
CVLT-List A	(*N* = 1230)	(*N* = 399)	(*N* = 428)	(*N* = 403)	0.14
	−0.85 ± 4.70	−1.26 ± 4.58	−0.56 ± 4.62 *	−0.76 ± 4.87	
CVLT-DFR	(*N* = 1210)	(*N* = 395)	(*N* = 421)	(*N* = 394)	0.031
	−0.24 ± 2.12	−0.40 ± 1.95	−0.24 ± 2.24	−0.07 ± 2.15 *	
BVRT	(*N* = 1241)	(*N* = 405)	(*N* = 436)	(*N* = 400)	<0.001
	+0.07 ± 4.39	+0.69 ± 4.55	+0.04 ± 4.41 *	−0.52 ± 4.12 ***^,f^	
BTA	(*N* = 1229)	(*N* = 398)	(*N* = 431)	(*N* = 400)	0.003
	−0.02 ± 1.41	−0.17 ± 1.43	−0.02 ± 1.42	+0.12 ± 1.37 **^,f^	
AF	(*N* = 1251)	(*N* = 406)	(*N* = 440)	(*N* = 405)	0.003
	+0.19 ± 4.22	−0.15 ± 3.84	−0.02 ± 4.2	+0.75 ± 4.53 **^,f^	
DS-F	(*N* = 1248)	(*N* = 406)	(*N* = 437)	(*N* = 405)	0.008
	+0.04 ± 1.64	−0.10 ± 1.64	+0.03 ± 1.60	+0.20 ± 1.67 **^,f^	
DS-B	(*N* = 1249)	(*N* = 406)	(*N* = 438)	(*N* = 405)	0.002
	+0.06 ± 1.61	−0.12 ± 1.53	+0.08 ± 1.60	+0.23 ± 1.68 **^,f^	
CDT	(*N* = 1248)	(*N* = 405)	(*N* = 440)	(*N* = 403)	0.016
	+0.01 ± 0.61	−0.07 ± 0.63	+0.03 ± 0.60 *	+0.04 ± 0.59 *	
TRAILS A	(*N* = 1235)	(*N* = 400)	(*N* = 434)	(*N* = 401)	0.13
	+0.10 ± 12.0	+0.68 ± 13.0	+0.23 ± 11.9	−0.62 ± 11.04	
TRAILS B	(*N* = 1224)	(*N* = 395)	(*N* = 432)	(*N* = 397)	0.33
	−6.59 ± 134.7	−4.14 ± 132.0	−2.56 ± 134.7	−13.4 ± 137.4	

Abbreviations: AF = Animal Fluency; BTA = Brief Test of Attention; BVRT = Benton Visual Retention Test; CDT = Clock Drawing Test; CES-D = Center for Epidemiologic Studies-Depression; CVLT-DFR = California Verbal Learning Test-Delayed Free Recall; CVLT-List A = California Verbal Learning Test-List A; DS-B = Digits Span-Backward; DS-F = Digits Span-Forward; HANDLS = Healthy Aging in Neighborhood of Diversity across the Lifespan; HS = High school; MMSE = Mini-Mental State Examination; *N*−3 = Omega-3; *N*-6 = Omega-6; PIR = Poverty income ratio; PUFA = Polyunsaturated Fatty Acids; RE = Retinol Equivalent; SD = Standard Deviation; T = tertile; TRAILS A = Trailmaking Test, Part A; TRAILS B = Trailmaking Test, part B; WRAT-3 = Wide Range Achievement Test, 3rd revision; *X* = mean. ^a^ Values are means (*X*) ± SD for continuous variables and % for categorical variables. The sample selected has complete data on MMSE at visits 1 and/or 2, complete exposure and covariate data for most study characteristics. For cognitive test performance score and change variables, test-specific samples were selected (Figure 1, Samples 3a–3k). Dietary total carotenoids was estimated as average of 2 24-h recalls from HANDLS first-visit (v1: 2004–2009). The measure summed across five main carotenoid groups, namely alpha-carotene, beta-carotene, beta-cryptoxanthin, lutein + zeaxanthin, and lycopene. The same visit and approach were applied to other dietary factors, including other antioxidants. Tertiles of total carotenoids were determined using the final analytic sample. All cognitive are in the direction of higher score → better performance with the exception of BVRT (# of errors) and TRAILS A and B (# of sec. to complete). ^b^
*p*-value from one-way analysis of variance (ANOVA, continuous variables) or from χ^2^ test (categorical variables). ^c^ Hypertension, diabetes, and dyslipidemia (or statin use) are based on clinical criteria from measured blood pressure, glucose, and total cholesterol, respectively, as well as self-report and related medication use at visit 1. Self-reported history of cardiovascular disease included atrial fibrillation, angina, coronary artery disease, congestive heart failure, or myocardial infarction. ^d^ Raw scores at v_1_ in the samples with complete and valid cognitive test scores at either visit, covariate and dietary data at v1. ^e^ Empirical bayes estimator of annual rate of change in each cognitive test score (mixed-effects linear regression models with TIME variable as the only predictor). ^f^
*p* < 0.05 upon further adjustment for age, sex, race, and poverty status in multiple linear, logistic, and multinomial logit models with carotenoid tertile entered as an ordinal variable. * *p* < 0.05, ** *p* < 0.01, *** *p* < 0.001, *t*-test for null hypothesis of no between-tertile differences, taking T_1_ as the referent.

**Table 2 nutrients-12-03558-t002:** Associations (unstandardized regression coefficients) of dietary vitamins A, C, and E with cognitive performance and decline over time, across total carotenoid tertiles: Mixed-effects linear regression models, HANDLS 2004–2013 ^a,b^.

	Exposure = Dietary Antioxidant					
	Vitamin A, per 1000 RE		Vitamin C, mg		Vitamin E, mg	
	Model 1	Model 2	Model 1	Model 2	Model 1	Model 2
	*γ* ± SE	*γ* ± SE	*γ* ± SE	*γ* ± SE	*γ* ± SE	*γ* ± SE
**Bottom tertile of total carotenoids**						
Outcome = Cognitive performance test score						
MMSE (*N* = 402, *k* = 1.9)						
Exposure, *γ*_0*a*_	−0.17 ± 0.15	−0.19± 0.15	+0.0005 ± 0.0023	−0.0008 ± 0.0023	+0.003 ± 0.035	−0.014 ± 0.036
Exposure × TIME, *γ*_1*a*_	+0.05 ± 0.03	+0.05 ± 0.03	+0.0005 ± 0.0004	+0.0007± 0.0005	+0.004 ± 0.008	+0.006 ± 0.008
CVLT-List A (*N* = 399, *k* = 1.8)						
Exposure, *γ*_0*a*_	+0.17 ± 0.45	+0.20 ± 0.45	+0.0004 ± 0.0075	+0.0001 ± 0.0076	+0.149 ± 0.118	+0.202 ± 0.121
Exposure × TIME, *γ*_1*a*_	−0.08 ± 0.10	−0.07 ± 0.09	+0.0014 ± 0.0016	+0.0018 ± 0.002	−0.004 ± 0.024	−0.013 ± 0.026
CVLT-DFR (*N* = 395, *k* = 1.7)						
Exposure, *γ*_0*a*_	+0.111 ± 0.211	+0.079 ± 0.211	+0.001 ± 0.003	+0.001 ± 0.003	+0.007 ± 0.053	+0.019 ± 0.055
Exposure × TIME, *γ*_1*a*_	−0.036 ± 0.051	−0.016 ± 0.052	−0.0006 ± 0.0008	−0.0003 ± 0.0008	+0.006 ± 0.011	+0.009 ±0.012
BVRT (*N* = 405, *k* = 1.9)						
Exposure, *γ*_0*a*_	+0.089 ± 0.32	+0.18 ± 0.32	−0.0017 ± 0.0062	−0.0019 ± 0.0063	−0.139 ± 0.098	−0.144 ± 0.102
Exposure × TIME, *γ*_1*a*_	−0.052 ± 0.080	−0.047 ± 0.077	−0.0006 ± 0.0013	−0.0000 ± 0.0013	+0.019 ± 0.020	+0.017 ± 0.020
BTA (*N* = 398, *k* = 1.8)						
Exposure, *γ*_0*a*_	−0.128 ± 0.132	−0.073 ± 0.134	+0.006 ± 0.003 *	+0.006 ±0.003 *	−0.038 ± 0.041	+0.0026 ± 0.043
Exposure × TIME, *γ*_1*a*_	−0.023 ± 0.038	−0.022 ± 0.038	+0.0001 ± 0.0006	+0.0001 ±0.0006	+0.004 ± 0.011	+0.003 ±0.043
AF (*N* = 406, *k* = 1.9)						
Exposure, *γ*_0*a*_	+0.032 ± 0.292	−0.000 ± 0.295	+0.005 ± 0.006	+0.002 ± 0.006	−0.082 ± 0.090	−0.118 ± 0.092
Exposure × TIME, *γ*_1*a*_	−0.020 ± 0.054	−0.019 ± 0.055	+0.001 ± 0.001	+0.001 ± 0.001	−0.012 ± 0.017	−0.001 ± 0.017
DS-F (*N* = 406, 1.8)						
Exposure, *γ*_0*a*_	+0.032 ± 0.029	−0.035 ± 0.127	+0.005 ± 0.006	+0.0002 ± 0.002	−0.082 ± 0.090	−0.069 ± 0.040
Exposure × TIME, *γ*_1*a*_	−0.020 ± 0.055	−0.021 ± 0.022	+0.001 ± 0.001	+0.000 ± 0.000	−0.012 ± 0.017	+0.001 ± 0.007
DS-B (*N* = 406, *k* = 1.8)						
Exposure, *γ*_0*a*_	−0.076 ± 0.112	−0.075 ± 0.112	+0.002 ± 0.002	+0.002 ± 0.002	−0.023 ± 0.034	−0.010 ± 0.035
Exposure × TIME, *γ*_1*a*_	−0.0003 ± 0.023	−0.0003 ±0.0227	−0.0018 ± 0.0004 *	−0.0011 ± 0.0005 *	+0.005 ± 0.008	−0.001 ± 0.009
CDT (*N* = 405, *k* = 2.0)						
Exposure, *γ*_0*a*_	−0.098 ± 0.094	−0.121 ± 0.094	+0.0009 ± 0.0014	+0.0002 ± 0.0015	+0.018 ± 0.022	+0.013 ± 0.023
Exposure × TIME, *γ*_1*a*_	+0.024 ± 0.021	+0.030 ± 0.022	−0.0002 ± 0.0003	−0.0002 ± 0.0004	−0.002 ± 0.005	−0.0014 ± 0.0056
TRAILS A (*N* = 400, *k* = 1.9)						
Exposure, *γ*_0*a*_	−1.70 ± 2.68	−1.70 ± 2.70	+0.023 ± 0.042	+0.035 ± 0.043	+0.076 ± 0.673	−0.223 ± 0.695
Exposure × TIME, *γ*_1*a*_	+0.135 ± 0.864	−0.078 ±0.868	−0.013 ± 0.012	−0.017 ± 0.012	−0.047 ± 0.191	−0.091 ± 0.197
TRAILS B (*N* = 395, *k* = 1.8)						
Exposure, *γ*_0*a*_	+7.6 ± 10.0	−10.8 ± 10.0	+0.244 ± 0.153	+0.172 ± 0.155	+6.00 ± 2.42 *	+4.15 ± 2.48
Exposure × TIME, *γ*_1*a*_	+0.63 ± 1.70	−0.16 ± 1.70	−0.035 ± 0.024	−0.028± 0.024	−0.10 ± 0.37	+0.06 ± 0.39
**Middle tertile of total carotenoids**						
Outcome = Cognitive performance test score						
MMSE (*N* = 438, *k* = 1.9)						
Exposure, *γ*_0*a*_	+0.024 ± 0.075	+0.034 ± 0.075	−0.002 ± 0.001	−0.002 ± 0.001 *	+0.020 ± 0.024	+0.019 ± 0.025
Exposure × TIME, *γ*_1*a*_	−0.015 ± 0.015 ^c^	−0.011 ± 0.015 ^c^	+0.0003 ± 0.0002	+0.0003 ± 0.0003	−0.0025 ± 0.0062	−0.0031 ± 0.0066
CVLT-List A (*N* = 429, *k* = 1.8)						
Exposure, *γ*_0*a*_	+0.090 ± 0.256	+0.101 ± 0.250	+0.0013 ± 0.0037	−0.0026 ± 0.0037	−0.0581 ± 0.0836 ^c^	−0.0978 ± 0.0856 ^c^
Exposure × TIME, *γ*_1*a*_	+0.083 ± 0.049	+0.088 ± 0.0483	−0.0000 ± 0.0008	+0.0004 ± 0.0008	+0.0112 ± 0.0234	+0.0207 ± 0.0251
CVLT-DFR (*N* = 421, *k* = 1.7)						
Exposure, *γ*_0*a*_	+0.165 ± 0.128	+0.198 ± 0.126	−0.001 ± 0.002	−0.0025 ± 0.0019	+0.0008 ± 0.0423	−0.0070 ± 0.0434
Exposure × TIME, *γ*_1*a*_	+0.016 ± 0.022	+0.0127 ± 0.022	+0.0003 ± 0.0004	+0.0003 ± 0.0004	+0.0018 ± 0.0102	+0.0001 ± 0.0108
BVRT (*N* = 436, *k* = 1.9)						
Exposure, *γ*_0*a*_	−0.223 ± 0.207	−0.2087 ±0.2043	+0.0021 ± 0.0028 ^c^	+0.0034 ± 0.0029	+0.1019 ± 0.0659 ^c^	+0.1030 ± 0.0685 ^c^
Exposure × TIME, *γ*_1*a*_	+0.013 ± 0.041	+0.010 ± 0.041	−0.0004 ± 0.0006	−0.0004 ± 0.0006	−0.0144 ± 0.0157	−0.0061 ± 0.0167
BTA (*N* = 431, *k* = 1.8)						
Exposure, *γ*_0*a*_	−0.049 ± 0.090	−0.0449± 0.0874	−0.0002 ± 0.0013 ^c^	−0.0007 ± 0.0013 ^c^	−0.0417 ± 0.0290	−0.0302 ± 0.0297
Exposure × TIME, *γ*_1*a*_	+0.000 ± 0.018	+0.000 ± 0.017	+0.0003 ± 0.0003	+0.0003 ± 0.0003	+0.0016 ± 0.0067	+0.002 ± 0.007
AF (*N* = 440, *k* = 2.0)						
Exposure, *γ*_0*a*_	−0.214 ± 0.207	−0. 193 ± 0.205	−0.0019 ± 0.0029	−0.0021 ±0.0029	+0.106 ± 0.065	+0.073 ±0.069
Exposure × TIME, *γ*_1*a*_	+0.027± 0.036	+0.0324 ± 0.036	+0.0000± 0.0005	+0.0001 ± 0.0006	−0.003 ± 0.013	+0.004 ± 0.014
DS-F (*N* = 437, *k* = 1.8)						
Exposure, *γ*_0*a*_	+0.0030 ± 0.0856	−0.0003 ± 0.0848	−0.0008 ± 0.0012	−0.0010 ± 0.0012	+0.064 ± 0.027 *^,c^	+0.053 ± 0.028 ^c^
Exposure × TIME, *γ*_1*a*_	+0.0200 ± 0.0157	+0.0195 ± 0.0154	+0.0001 ± 0.0002	+0.0001± 0.0003	+0.0022 ± 0.0069	−0.00027 ± 0.0073
DS-B (*N* = 438, *k* = 1.8)						
Exposure, *γ*_0*a*_	+0.080 ± 0.080	+0.087 ± 0.080	+0.001 ± 0.001	+0.001 ± 0.001	−0.026 ± 0.025	−0.041 ± 0.026
Exposure × TIME, *γ*_1*a*_	+0.014 ± 0.015	+0.014 ±0.015	−0.0003 ±0.0002	−0.0003 ±0.0002	+0.0130 ± 0.0063 *	+0.0130 ± 0.0068
CDT (*N* = 440, *k* = 2.0)						
Exposure, *γ*_0*a*_	−0.038 ± 0.054	−0.035 ± 0.053	+0.0002 ± 0.0007	−0.0001 ± 0.0007	+0.0171 ± 0.0168	+0.0149 ±0.0176
Exposure × TIME, *γ*_1*a*_	+0.009 ± 0.012	+0.008 ± 0.012	+0.0002 ± 0.0002	+0.0003 ± 0.0002	−0.0022 ± 0.0042	−0.0015 ± 0.0044
TRAILS A (*N* = 434, *k* = 1.9)						
Exposure, *γ*_0*a*_	−0.349 ± 1.578	−0.654 ±1.580	−0.0162 ± 0.0216	−0.0095 ± 0.022	−0.2302 ± 0.4934	−0.2137 ± 0.5209
Exposure × TIME, *γ*_1*a*_	+0.065 ± 0.429	+0.097 ± 0.431	+0.0023 ± 0.0066	−0.0019 ± 0.0070 ^c^	−0.0250 ± 0.1646	−0.1108 ± 0.1786
TRAILS B (*N* = 432, *k* = 1.9)						
Exposure, *γ*_0*a*_	+3.690 ± 5.810	3.144 ± 5.725	+0.037 ± 0.080	+0.069 ± 0.082 ^c^	+2.610 ± 1.841	+2.63 ±1.91
Exposure × TIME, *γ*_1*a*_	+0.099 ± 1.191	−0.028 ± 1.197	−0.028± 0.018	−0.025 ±0.019	**−1.250 ± 0.470 ****	−1.19 ± 0.50 *
**Top tertile of total carotenoids**						
Outcome = Cognitive performance test score						
MMSE (*N* = 401, *k* = 1.9)						
Exposure, *γ*_0*a*_	+0.049 ± 0.114	+0.150 ± 0.119	−0.0014 ± 0.0010	−0.0009 ± 0.0010	+0.0167 ± 0.0183	+0.027 ± 0.020
Exposure × TIME, *γ*_1*a*_	−0.021 ± 0.025	−0.045 ± 0.026	+0.0003 ± 0.0002	+0.0001 ± 0.0002	+0.0040 ± 0.0040	−0.001 ± 0.005
CVLT-List A (*N* = 403, *k* = 1.7)						
Exposure, *γ*_0*a*_	+0.146 ± 0.423	+0.236 ± 0.441	+0.0060 ± 0.0037	+0.0065 ± 0.0038	**+0.259 ± 0.082 ****	+0.221± 0.089 *
Exposure × TIME, *γ*_1*a*_	−0.047 ± 0.079	−0.074 ± 0.083	−0.001 ± 0.001	−0.0010 ± 0.0008	−0.021 ± 0.0171	−0.023 ± 0.0185
CVLT-DFR (*N* = 394, *k* = 1.7)						
Exposure, *γ*_0*a*_	+0.2581 ±0.3174	+0.303± 0.334	+0.001 ± 0.002	+0.0009 ± 0.0017	+0.076 ± 0.038 *	+0.064 ± 0.041
Exposure × TIME, *γ*_1*a*_	−0.085 ± 0.066	−0.102 ± 0.069	+0.0001 ± 0.0004	+0.0001 ± 0.0004	−0.0115 ± 0.009	−0.013 ± 0.009
BVRT (*N* = 400, *k* = 1.9)						
Exposure, *γ*_0*a*_	−0.607 ± 0.318	−0.695 ± 0.334 *	−0.0012 ± 0.0027	−0.0012 ± 0.0028	−0.096 ± 0.051	−0.098 ± 0.055
Exposure × TIME, *γ*_1*a*_	**+0.280 ± 0.065 ***^,c^**	**+0.272 ± 0.068 ***^,c^**	+0.0003 ± 0.0006	+0.0005 ± 0.0006	−0.0035 ± 0.0109	+0.0003 ± 0.0120
BTA (*N* = 400, *k* = 1.8)						
Exposure, *γ*_0*a*_	+0.202 ± 0.228	+0.152 ± 0.236	+0.0003 ± 0.00130	+0.0003 ± 0.0013 ^c^	+0.0141 ± 0.0240	+0.0056 ± 0.0254
Exposure × TIME, *γ*_1*a*_	−0.035 ± 0.046	+0.037 ± 0.047	+0.00023 ± 0.00028	+0.0004 ± 0.0003	−0.0055 ± 0.0052	−0.0015 ± 0.0055
AF (*N* = 405, *k* = 2.0)						
Exposure, *γ*_0*a*_	+0.184 ± 0.386	+0.206 ± 0.401	+0.0035 ± 0.0032	0.0016 ± 0.0034	+0.118 ± 0.062	+0.0741 ± 0.0665
Exposure × TIME, *γ*_1*a*_	+0.010 ±0.063	+0.003 ± 0.067	−0.0010 ±0.0006	−0.0010 ± 0.0006	+0.0016 ± 0.0109	+0.0003 ± 0.0119
DS-F (*N* = 405, *k* = 1.9)						
Exposure, *γ*_0*a*_	−0.031 ± 0.203	−0.218 ± 0.212	−0.0000 ± 0.0012	−0.0001 ± 0.0012	+0.0253 ± 0.0222 ^c^	+0.0271± 0.0235 ^c^
Exposure × TIME, *γ*_1*a*_	+0.050 ± 0.039	+0.070 ± 0.041	−0.0001 ± 0.0002	−0.0001 ± 0.0002	+0.0037 ± 0.0041	+0.0270 ± 0.02355
DS-B (*N* = 405, *k* = 1.9)						
Exposure, *γ*_0*a*_	+0.081 ± 0.209	+0.247 ± 0.219	+0.0008 ± 0.0012	+0.0007 ± 0.0012	+0.0471 ± 0.0220 *^,c^	+0.0448 ± 0.0236
Exposure × TIME, *γ*_1*a*_	+0.011 ± 0.041	−0.029 ± 0.043	−0.0001 ± 0.0002	−0.0001 ± 0.0002	+0.0008 ± 0.0045	−0.0006 ± 0.0048
CDT (*N* = 403, *k* = 2.0)						
Exposure, *γ*_0*a*_	−0.0843 ± 0.0080	−0.0782 ± 0.0833	+0.0012 ± 0.0007	+0.0012 ± 0.0007	+0.0056 ± 0.0129	+0.0058 ± 0.0139
Exposure × TIME, *γ*_1*a*_	+0.0018 ± 0.0182	−0.0000 ± 0.0192	−0.0002 ± 0.0002	−0.00019 ± 0.00017	+0.0052 ± 0.0032	+0.0062 ± 0.0034
TRAILS A (*N* = 401, *k* = 1.9)						
Exposure, *γ*_0*a*_	−1.259 ± 2.435	−2.035 ± 2.551	−0.0118 ± 0.0211	−0.00589 ± 0.0219	+0.1482 ± 0.3940	+0.4112 ± 0.4284
Exposure × TIME, *γ*_1*a*_	+0.651 ± 0.592	+0.721 ± 0.621	+0.0001 ± 0.0055	−0.00207 ± 0.00568 ^c^	−0.0813 ± 0.1034	−0.1243 ± 0.1123
TRAILS B (*N* = 397, *k* = 1.9)						
Exposure, *γ*_0*a*_	+16.15 ± 9.92	+11.06 ± 10.29	+0.0873 ± 0.0856	+0.0410 ± 0.0880 ^c^	−0.017 ± 1.595 ^c^	+0.1139 ± 1.7165 ^c^
Exposure × TIME, *γ*_1*a*_	−0.194 ± 2.574	−0.006 ± 2.717	+0.0033 ± 0.0162	+0.0085 ± 0.0166	−0.439 ± 0.279	−0.4481 ± 0.3018

Abbreviations: *γ* = Fixed effect; AF = Animal Fluency; BTA = Brief Test of Attention; BVRT = Benton Visual Retention Test; CDT = Clock Drawing Test; CES-D = Center for Epidemiologic Studies-Depression; CVLT-DFR = California Verbal Learning Test-Delayed Free Recall; CVLT-List A = California Verbal Learning Test-List A; DS-B = Digits Span-Backward; DS-F = Digits Span-Forward; HANDLS = Healthy Aging in Neighborhood of Diversity across the Lifespan; HS = High school; MMSE = Mini-Mental State Examination; *N*-3 = Omega-3; *N*-6 = Omega-6; PIR = Poverty income ratio; PUFA = Polyunsaturated Fatty Acids; SE = standard error; T = tertile; TRAILS A = Trailmaking Test, Part A; TRAILS B = Trailmaking Test, part B; WRAT-3 = Wide Range Achievement Test, 3rd revision. ^a^ The sample selected has complete data on each of the cognitive test scores at visits 1 (v1: 2004–2009) and/or 2 (v2: 2009–2013) and complete data on exposures and covariates at v1. Vitamins A, C, and E were estimated at v1 as average of two 24-h recalls from HANDLS. The same approach was applied to total dietary carotenoid intake and other dietary variables. All cognitive are in the direction of higher score → better performance with the exception of BVRT (# of errors) and TRAILS A and B (# of sec. to complete). ^b^ Mixed-effects linear regression models were conducted with each cognitive test score as the outcome and each of dietary vitamin A, C, and E as exposures (See Appendix A for notations). Random effects were added to the intercept and TIME. Exposures and covariates were interacted with TIME. Minimally adjusted models (Model 1) included age at v1 (in years) centered at 48, sex (male vs. female), race (African-American vs. White), poverty status (below vs. above poverty), education (<HS, HS, >HS), WRAT-3 total score (centered at 42) and employment status (unemployed vs. employed, missing vs. employed), total energy intake (kcal/d, centered at 2006), total carotenoids (centered at 7478), the other two antioxidants and the inverse mills ratio. Model 2 is Model 1, further adjusted for tobacco and drug use, HEI-2010 total score (centered at 43), BMI (centered at 30), self-rated health, CES-D total score (centered at 15), hypertension, type 2 diabetes, dyslipidemia (or statin use), cardiovascular disease, and supplement use (visits 1 and 2 vs. non-user; visit 2 only vs. non-user). Models are run stratifying by tertile of total dietary carotenoid intake. All cognitive test scores are in the direction of higher score → better performance with the exception of BVRT (# of errors) and TRAILS A and B (# of sec. to complete). See Table 1 for tertile distribution of total carotenoids. * *p* < 0.05, ** *p* < 0.01, *** *p* < 0.001, for null hypothesis that *γ* = 0. Bolded values passed correction for multiple testing. ^c^
*p* < 0.10 for null hypothesis of that *γ* = 0 in unstratified models with interaction between tertiles of dietary total carotenoid intake and each of the 3 vitamin exposures, taking T_1_ as the referent category.

**Table 3 nutrients-12-03558-t003:** Interactions between individual carotenoids and other antioxidants (vitamins A, E, and C) in relation to v1 cognitive test score and annual rate of change in cognitive performance: Summary of findings with *p* < 0.05. ^a,b^

	Carotenoid				
	Alpha-Carotene	Beta-Carotene	Lutein + Zeaxanthin	Beta-Cryptoxanthin	Lycopene
**Vitamin A**					
Effect on v1 Cognitive test score: Vitamin A × Carotenoid, *γ*_0*a*_	CVLT-List A (+); AF (+); TRAILS B (−)	BVRT (−)	--	--	--
Effect on Cognitive test score change: Vitamin A × Carotenoid × Time, *γ*_1*a*_	--	**BVRT (+)**; CDT (−)	**BVRT(+)**; TRAILS B (−)	DS-F (+); CDT (−)	--
**Vitamin C**					
Effect on v1 Cognitive test score: Vitamin C × Carotenoid, *γ*_0*a*_	CVLT-List A (+); TRAILS B (−)	CVLT-List A (+)	--	--	--
Effect on Cognitive test score change: Vitamin C × Carotenoid × Time, *γ*_1*a*_	CDT (−)	CDT (−)	CDT (−)	MMSE (−)	--
**Vitamin E**					
Effect on v1 Cognitive test score: Vitamin E × Carotenoid, *γ*_0*a*_	AF (+); DS-F (+); **DS-B (+)**; TRAILS B (−)	DS-B (+)	--	--	CVLT-List A (+)
Effect on Cognitive test score change: Vitamin E × Carotenoid × Time, *γ*_1*a*_	--	--	--	--	--

Abbreviations: *γ* = Fixed effect; AF = Animal Fluency; BTA = Brief Test of Attention; BVRT = Benton Visual Retention Test; CDT = Clock Drawing Test; CES-D = Center for Epidemiologic Studies-Depression; CVLT-DFR = California Verbal Learning Test-Delayed Free Recall; CVLT-List A = California Verbal Learning Test-List A; DS-B = Digits Span-Backward; DS-F = Digits Span-Forward; HANDLS = Healthy Aging in Neighborhood of Diversity across the Lifespan; HS = High school; MMSE = Mini-Mental State Examination; *N*-3 = Omega-3; *N*-6 = Omega-6; PIR = Poverty income ratio; PUFA = Polyunsaturated Fatty Acids; SE = standard error; T = tertile; TRAILS A = Trailmaking Test, Part A; TRAILS B = Trailmaking Test, part B; WRAT-3 = Wide Range Achievement Test, 3rd revision. ^a^ The sample selected has complete data on each of the cognitive test scores at visits 1 and/or 2, complete data on exposures and covariates. Vitamins A, C, and E were estimated as average of 2 24-h recalls from HANDLS first-visit (v1: 2004–2009). The same approach was applied to total and individual dietary carotenoid intakes, as well as other dietary variables. All cognitive are in the direction of higher score → better performance with the exception of BVRT (# of errors) and TRAILS A and B (# of sec. to complete). ^b^ Mixed-effects linear regression models were conducted with each cognitive test score as the outcome and each of dietary vitamin A, C, and E as exposures. Random effects were added to the intercept and TIME (See Appendix A for notations). Exposures and covariates were interacted with TIME. Minimally adjusted models (Model 1) included age at v1 (in years) centered at 48, sex (male vs. female), race (African-American vs. White), poverty status (below vs. above poverty), education (<HS, HS, >HS), WRAT-3 total score (centered at 42), and employment status (unemployed vs. employed, missing vs. employed), total energy intake (kcal/day, centered at 2006), total carotenoids (centered at 7478), the other two antioxidants and the inverse mills ratio. Further adjustment was also made on tobacco and drug use, HEI-2010 total score (centered at 43), BMI (centered at 30), self-rated health, CES-D total score (centered at 15), hypertension, type 2 diabetes, dyslipidemia (or statin use), cardiovascular disease, and supplement use (visits 1 and 2 vs. non-user; visit 2 only vs. non-user). Further, in separate models, each of the three antioxidants (vitamins A, C, and E) were interacted with each of the five individual carotenoids. Findings are summarized in terms of significant interactions for a specific cognitive test score, at type I error of 0.05 and the direction of the parameter point estimate: +: positive; −: negative. Bolded findings are associated with a *p* < 0.001. All cognitive test scores are in the direction of higher score → better performance with the exception of BVRT (# of errors) and TRAILS A and B (# of sec. to complete). See Table 1 for tertile distribution of total carotenoids.

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
