# Peer review of "Association of Antioxidant Vitamins A, C, E and Carotenoids with Cognitive Performance over Time: A Cohort Study of Middle-Aged Adults"

_nutrients, 2020, doi:10.3390/nu12113558_

Round 1
Reviewer 1 Report
Review
In the manuscript “Association of antioxidant vitamins A, C, E and carotenoids with cognitive performance over time: a cohort study of middle-aged adults" the authors explore the relationship between several dietary antioxidants and cognitive performance and change over time.
The topic is interesting and also the size of the population is adequate. I find the statistics well describe and correctly used. Thank you for giving me the opportunity to revise it. I hope my criticism will help to improve the manuscript.
I would like to highlight a few points that should be changed or added.
- There is no justification why studies of cognitive function were performed in such young people.
- The manuscript contains a large amount of information, making it difficult to read and follow the author's line of thought. The tables are extensive and therefore difficult to analyze. Perhaps it would be worth redrafting or separating them. It would be worth presenting only the most relevant information or highlighting it. You can also think about the chart. Please consider transferring some of the results to a supplement
- In section 2.5. Cognitive Outcome Measures a bibliography should be added.
- The title of section 2.4 is duplicated (Line 132 and 146).
- Line 204: Stata release 16, usually name of this software is written with a capital letter Stata Release 16.
- In Conclusion (Line 518-520) is: “The clinical interpretation and implications of this study are that diet supplementation with vitamin E and carotenoids, including lycopene, may reduce the likelihood of cognitive decrements in the short term, particularly in the domain of verbal memory.” But the analyses concentrate on dietary intake not on supplementation. I suggest to change to “diet rich in ….”
- The bibliography is extensive and formats properly.
- English is correct and really nice to read.
Author Response
Reviewer 1:
In the manuscript “Association of antioxidant vitamins A, C, E and carotenoids with cognitive performance over time: a cohort study of middle-aged adults" the authors explore the relationship between several dietary antioxidants and cognitive performance and change over time.
The topic is interesting and also the size of the population is adequate. I find the statistics well describe and correctly used. Thank you for giving me the opportunity to revise it. I hope my criticism will help to improve the manuscript.
I would like to highlight a few points that should be changed or added.
- There is no justification why studies of cognitive function were performed in such young people.
Response: The following sentence was added, based on a previously published study: “Given that sub-clinical cognitive deterioration in older age may impact dietary intakes at baseline, it is thus crucial to examine associations between dietary antioxidant and cognitive outcomes, including change over time, in a relatively young and healthy population[1].”
- The manuscript contains a large amount of information, making it difficult to read and follow the author's line of thought. The tables are extensive and therefore difficult to analyze. Perhaps it would be worth redrafting or separating them. It would be worth presenting only the most relevant information or highlighting it. You can also think about the chart. Please consider transferring some of the results to a supplement
Response: We appreciate this comment. We have reduced the description in Table 1 (see changes highlighted in yellow), mostly in the first part of the paragraph:
“Study sample characteristics are presented in Table 1 across v1 dietary carotenoid tertiles. Upward linear trends across tertiles of carotenoids were observed for most dietary factors (i.e. individual carotenoids, vitamins A, C and E, total caloric intake and HEI-2010). Moreover, the uppermost tertile of dietary carotenoids (T3) compared with the lowest tertile (T1) had a higher socio-economic status, in terms of education and employment status, as well as elevations in mean WRAT-3 score and proportion with very good/excellent health, while having a lower proportion of African-Americans, current tobacco users, individuals with hypertension and/or diabetes.”
We have also removed the detailed sample sizes prior to the final sample in Figure 1. Abbreviations and a title were added.
- In section 2.5. Cognitive Outcome Measures a bibliography should be added.
Response: Done. The references from the supplemental materials were included.
- The title of section 2.4 is duplicated (Line 132 and 146).
Response: The second title was removed. Thank you!
- Line 204: Stata release 16, usually name of this software is written with a capital letter Stata Release 16.
Response: Done.
- In Conclusion (Line 518-520) is: “The clinical interpretation and implications of this study are that diet supplementation with vitamin E and carotenoids, including lycopene, may reduce the likelihood of cognitive decrements in the short term, particularly in the domain of verbal memory.” But the analyses concentrate on dietary intake not on supplementation. I suggest to change to “diet rich in ….”
Response: Done.
- The bibliography is extensive and formats properly.
Response: Thank you!
- English is correct and really nice to read.
Response: Thank you!
Reviewer 2 Report
Beydoun et al. presented partial evidence for synergism between vitamin E and carotenoids in relation to better baseline cognitive performance. This is an interesting prospective cohort study, making use of several cognitive function tests, advanced statistical techniques such as multiple linear mixed-effects regression models. There are, however, several issues to be addressed to further improve the manuscript.
- Although the authors employed data from 24-hr recalls as an estimate of usual dietary intake of antioxidants, scientific credibility of the obtained results should be poor due to lack of plasma concentration of antioxidants.
- I understand the lack of factorial invariance by race, sex, or poverty status groups in the present study. However, according to the body mass index, almost all participants were obese. Isn’t there a serious bias in the study population?
Author Response
Reviewer 2:
Beydoun et al. presented partial evidence for synergism between vitamin E and carotenoids in relation to better baseline cognitive performance. This is an interesting prospective cohort study, making use of several cognitive function tests, advanced statistical techniques such as multiple linear mixed-effects regression models. There are, however, several issues to be addressed to further improve the manuscript.
- Although the authors employed data from 24-hr recalls as an estimate of usual dietary intake of antioxidants, scientific credibility of the obtained results should be poor due to lack of plasma concentration of antioxidants.
Response: We agree this is a major limitation. However, at least a previous study using nationally representative data has shown a moderate correlation between each of the antioxidants and carotenoids in the diet with their corresponding plasma level values. This was added to the Discussion, limitations section as follows: “Nevertheless, several large studies of healthy adults have shown that dietary values of each of those antioxidants and carotenoids represented in our study measured with 2 24 hr recalls had moderate correlation with their respective plasma biomarker values[2,3].”
- I understand the lack of factorial invariance by race, sex, or poverty status groups in the present study. However, according to the body mass index, almost all participants were obese. Isn’t there a serious bias in the study population?
Response: We appreciate this comment. However, the mean BMI is 31, making about half of the sample obese, which is an overestimate of the national prevalence but a close enough estimate for the prevalence in urban socio-economically and racially diverse populations in the US. The external validity of the study was clarified as follows among strengths of the study: “Our findings can also be extrapolated to several African American and White populations, as HANDLS is representative of 14 urban settings across the United States.
- Nooyens, A.C.; Milder, I.E.; van Gelder, B.M.; Bueno-de-Mesquita, H.B.; van Boxtel, M.P.; Verschuren, W.M. Diet and cognitive decline at middle age: the role of antioxidants. Br J Nutr 2015, 113, 1410-1417, doi:10.1017/S0007114515000720.
- Zhao, Y.; Monahan, F.J.; McNulty, B.A.; Gibney, M.J.; Gibney, E.R. Effect of vitamin E intake from food and supplement sources on plasma alpha- and gamma-tocopherol concentrations in a healthy Irish adult population. Br J Nutr 2014, 112, 1575-1585, doi:10.1017/S0007114514002438.
- Yang, M.; Chung, S.J.; Floegel, A.; Song, W.O.; Koo, S.I.; Chun, O.K. Dietary antioxidant capacity is associated with improved serum antioxidant status and decreased serum C-reactive protein and plasma homocysteine concentrations. Eur J Nutr 2013, 52, 1901-1911, doi:10.1007/s00394-012-0491-5.